# Photodynamic Therapy Supported by Antitumor Lipids

**DOI:** 10.3390/pharmaceutics15122723

**Published:** 2023-12-03

**Authors:** Mladen Korbelik

**Affiliations:** Department of Integrative Oncology, BC Cancer, Vancouver, BC V5Z 1L3, Canada; mkorbelik@bccrc.ca; Tel.: +1-604-675-8084

**Keywords:** photodynamic therapy, antitumor lipids, edelfosine, mouse tumor models

## Abstract

Photodynamic therapy (PDT) destroys tumors by generating cytotoxic oxidants that induce oxidative stress in targeted cancer cells. Antitumor lipids developed for cancer therapy act also by inflicting similar stress. The present study investigated whether tumor response to PDT can be improved by adjuvant treatment with such lipids using the prototype molecule edelfosine. Cellular stress intensity following Photofrin-based PDT, edelfosine treatment, or their combination was assessed by the expression of heat shock protein 70 (HSP70) on the surface of treated SCCVII tumor cells by FITC-conjugated anti-HSP70 antibody staining and flow cytometry. Surface HSP70 levels that became elevated after either PDT or edelfosine rose much higher after their combined treatment. The impact of Photofrin-PDT-plus-edelfosine treatment was studied with three types of tumor models grown in syngeneic mice. With both SCCVII squamous cell carcinomas and MCA205 fibrosarcoma, the greatest impact was with edelfosine peritumoral injection at 24 h after PDT, which substantially improved tumor cure rates. With Lewis lung carcinomas, edelfosine was highly effective in elevating PDT-mediated tumor cure rates even when injected peritumorally immediately after PDT. Edelfosine used before PDT was ineffective as adjuvant with all tumor models. The study findings provide proof-in-principle for use of cancer lipids with tumor PDT.

## 1. Introduction

Photodynamic therapy (PDT) destroys tumors and other targeted lesions by inflicting oxidative stress using light treatment to activate photosensitizing drugs, generating cytotoxic oxidants including singlet molecular oxygen [1]. As a clinically verified approach, PDT is being used for treating a variety of malignant solid tumors [1,2] and diverse other conditions including wound healing [3]. Classical PDT operates by integrating the actions of three main components: photosensitizing agent, non-ionizing radiation (usually tissue-penetrating light from the red segment of the visible spectrum), and molecular oxygen [1]. The simultaneous engagement of all three components is necessary for activating the photodynamic mechanisms. Prevalent photodynamic reaction comprises energy transfer from a photoexcited triplet state of the photosensitizer to ground-state molecular oxygen (Type II process) [1,4]. This results in the formation of singlet molecular oxygen, which is a short-lived reactive oxygen species (ROS), and represents the most abundant directly formed cytotoxic oxygen species in PDT. Some photosensitizers can be excited into a relatively long-lived triplet state, which can allow a free radical-mediated Type I process with electron transfer to O_2,_ resulting in the formation of superoxide (O_2_^−^•) and additional free radical ROS [4]. Light delivery (typically through fiber optic cables) is focused on the target area (tumor), rendering PDT highly selective limiting damage to surrounding normal tissues, and this makes it more discriminatory than other conventional cancer therapies. In addition to the less invasive approach and excellent overall cosmetic outcome, the advantages of PDT include the absence of long-term side effects and possibility to be repeated many times due to absence of resistance mechanisms [2]. In case of a short time interval between photosensitizer administration and light treatment, the principal damage in the PDT-treated tumor is focused to its vasculature (hence vascular PDT), with tumor cell death caused by vasculature occlusion [1]. In addition to the direct killing of tumor cells by generated ROS and the indirect dying of these cells consequent to vascular damage, the third important mechanism of tumor destruction by PDT is by an elicited strong antitumor immune response [1,5]. Indeed, PDT is widely recognized as a modality suitable to extremely efficaciously engage the patient’s immune system through the induction of immunogenic cell death (ICD), release of damage-associated molecular patterns (DAMPs), and securing greatly increased availability of highly immunogenic tumor antigens [5]. This can result in systemic tumor immune rejection and immunological memory generation capable of preventing disease recurrence.

Phospholipids and cholesterol, as well as other unsaturated lipids in cell membranes, are subjected to oxidative modification by PDT-induced stress, predominantly manifested as lipid peroxidation [6,7]. Lipid hydroperoxides (LOOHs), including cholesterol hydroxyperoxides, are key non-radical intermediates of this peroxidative process. The inflicted peroxidative injury is associated with various pathological conditions and is a prominent contributor in the impact of PDT on tumors [6]. Importantly, membrane lipid damage is prone to spread to other vital compartments. The membrane-disruptive potential of LOOHs is made more powerful because of their much longer lifetime than their free radical precursors and singlet oxygen [7]. This allows them to migrate from the origin site to more sensitive sites, which also makes them more dangerous. Indeed, LOOHs appear to get engaged in secondary propagative reactions long after PDT challenge, resulting in damage that far exceeds that inflicted in the initial light-dependent phase [7]. Lipid hydroperoxides formed after PDT are also known participants in stress signaling cascades securing either antioxidant defense response or switching to cell death signaling [4,7].

The survival of PDT-treated tumor cells is largely dependent on cellular membrane repair mechanisms engaging SREBPs (sterol regulatory element-binding proteins) pathway. This signaling cascade mediates the control of cholesterol and fatty acid metabolism [8]. Peroxidative lipid injury caused by PDT disrupts lipid homeostasis, precipitating the fall in cellular levels of cholesterol and other lipids. This is detected by an ER-localized cholesterol-sensing protein named SREBP cleavage-activating protein (SCAP) [9]. The consequent conformational change in SCAP prompts its inclusion (and of escorted SCREBP) into the vesicles, transporting proteins from the ER to the Golgi apparatus. In the Golgi, site-1 protease (S1P) and site-2 protease (S2P) sequentially cleave SREBP, releasing the active NH_2_-terminal transcription factor domain, which allows the now-activated transcription factor to travel to the nucleus and activates genes instrumental for lipogenesis [9]. We showed earlier that treatment with the SREBP inhibitor fatostatin A strongly reduces the survival of PDT-treated SCCVII tumor cells [10]. Fatostatin A is diarylthiazole derivative that binds to SCAP and blocks the SCAP/SREBP complex translocation to the Golgi [11]. The present study was designed to investigate whether tumor resistance to PDT, influenced by this repair activity, could be largely hampered by antitumor lipids.

A major hallmark of cancer is metabolic deregulation and reprogramming [12]. Aberrant lipid metabolism in cancer cells is characterized by a shift towards increased lipogenesis, with reduced reliance on dietary lipids and liver-synthesized lipids [13,14]. Consequently, these cells have an increased lipid content and derive energetic substrates from lipid-dependent catabolism. This inspired the development of synthetic alkylphospholipid analogues and other membrane-disrupting antitumor lipids as anti-neoplastic drugs with established properties of selective uptake by tumor tissue, due to their accumulation in membrane structures and the endoplasmic reticulum (ER) of cancer cells [15]. Such agents affect lipid composition and cholesterol content and exhibit distinct antiproliferative properties selectively in tumor cells owing to their interference with lipid metabolism (particularly de novo phospholipid biosynthesis) and obstruction of lipid-dependent signaling [13]. The latter affects critical survival signaling pathways PI3K-Akt and Raf-Erk1/2 [15]. Another reported effect by these agents was the potent and persistent activation of protein kinase JNK responsible for the phosphorylation of c-Jun transcription factor, known as a component of AP-1 transcription factor complex and associated with its activation [16]. The plasma membrane appears to be the decisive target for the cytotoxic action of these agents, since they were found to affect the membrane permeability, fluidity, membrane lipid raft function, lipid composition and cholesterol content of these structures [16]. Lipid rafts are ordered membrane domains enriched in cholesterol and saturated lipids that influence membrane fluidity, membrane protein trafficking (including receptors clustering), recruit certain lipids and proteins into their structure, and are involved in the formation of endocytic, exocytic, synaptic and other vesicles [17]. The disruption of lipid raft composition by antitumor lipids on the plasma membrane and possibly mitochondrial membrane was suggested to modulate the distribution of death receptor Fas/CD95, enhancing their recruitment and leading to the activation of apoptosis of tumor cells in a ligand-dependent manner [16,18]. Indeed, Fas/CD95 and lipid raft clustering by alkylphospholipid agents was found to promote further recruitment of apoptotic signaling molecules into lipid rafts, culminating in the formation of the major apoptotic complex DISC [18].

Edelfosine was chosen as the antitumor lipid in this study, because it is considered the prototype molecule of its class and is one of the most investigated ether lipids [16]. Its general effect on cancer cells manifests as the disruption of cholesterol homeostasis, which is a consequence of the stimulation of cholesterol biosynthesis and downregulation of the main pathways for removal of the excess amounts of cholesterol/lipids from the cells [19].

Since antitumor lipids were proven to be therapeutically particularly effective in combination with radiotherapy and other anti-cancer agents in various pre-clinical and clinical studies [12], we investigated the prospects of improved tumor control in protocols combining PDT with antitumor lipid treatment. This was motivated by the fact that, similarly to antitumor lipids, PDT inflicts damage to cellular membrane structures and instigates ER stress [20,21].

## 2. Materials and Methods

### 2.1. Drugs

Edelfosine, (ET-19-O-CH3; 1-octadecyl-2-*O*-methyl-glycero-3-phosphocholine) was obtained from Calbiochem Merk KGaA (Darmstadt, Germany). It was dissolved in ethanol to form 10 mmol/mL (5.24 mg/mL) stock solution. This was diluted at least 200 times in cell growth medium for the final experimental concentration, with cells 48 μmol/mL (25 μg/mL). For in vivo, the injection volume contained edelfosine stock solution diluted in phosphate-buffered saline (PBS); for instance, the i.p. edelfosine injection of 0.1 mg/mouse (0.194 mmol/mouse) was performed with 19 μL of edelfosine stock combined with 200 μL PBS.

Photosensitizer Photofrin was obtained from Axcan Pharma (Mont-SaintHillaire, QC, Canada). From its original stock, it was diluted either in growth medium (in vitro experiments) or in physiological saline (in vivo treatments).

### 2.2. Animals

Cultured SCCVII cells that originated from murine squamous cell carcinoma [22] were grown in alpha minimal essential medium with 100 μg/mL streptomycin and 100 U/mL penicillin (Sigma Chemical Co., St. Louis, MO, USA) and also containing 10% fetal bovine serum (HyClone Laboratories, Inc., Logan, UT, USA). Their cultures were maintained with weekly fresh medium changes and were sub cultured once a week at a 1:50 ratio. Experimental tumors were inoculated using cell suspensions obtained by the enzymatic digestion of SCCVII tumor tissue. Cohorts of SCCVII tumors were implanted by inoculating 30 μL of PBS containing one million SCCVII cells and were injected s.c. into lower dorsal area of 7- to 9-week-old female syngeneic immunocompetent C3H/HeN mice. The use of anesthesia was avoided during this procedure by keeping the mice immobilized in lead holders. The SCCVII tumors are a recognized model of head and neck cancer of spontaneous origin. The same procedure was used for implanting fibrosarcomas MCA205 [23] and Lewis lung carcinomas [24] that were both grown in the same subcutaneous location in syngeneic immuno-non-compromised C57BL/6 mice. The tumors were treated upon reaching 7–8 mm as the largest diameter. The results depicted in Section 3 were obtained with each treatment group containing 8 mice. The mice were kept in Animal Resource Centre of BC Cancer Research Centre and housed under a controlled barrier environment. Drinking water and standard laboratory chow were present ad libitum. The experimental protocols/procedures with mice were performed within the guidelines of international and Canadian policies based on the Basel declaration and the 3R concept, with every effort made to avoid animal suffering and reduce the number of used animals. The project and operating procedures were approved by the Animal care committee of the University of British Columbia (project A05-0211).

### 2.3. Photodynamic Therapy

For in vitro PDT, Photofrin was incubated with cells for 24 h in complete growth medium at 20 μg/mL. The dishes with attached cells were rinsed with PBS, then the cells were detached by trypsinization and transferred to ice-cold PBS for exposure to a 630 nm ± 10 nm light dose of 1 J/cm^2^ (15 mW/cm^2^). For in vivo PDT, mice injected i.v. with Photofrin 24 h earlier had their tumors plus 1 mm surrounding area illuminated superficially with the same light at 110–120 mW/cm^2^, while they were restrained unanesthetized in holders designed to expose the lower region of their backs. The details about the used QTH lamp illuminator and the light delivery were described earlier [25]. Briefly, the light was generated by a 150 W QTH lamp-powered high-throughput source equipped with an integrated ellipsoid reflector and furnished with interchangeable interference filters (FB-QTH-3 model, manufactured by Sciencetech Inc., London, ON, Canada). Liquid light guide model 77638 (Oriel Instruments, Stratford, CT, USA) was used for light delivery into the target area. The mice were thereafter monitored daily with the recording of the presence/absence of tumor growth; the absence of a palpable tumor at 90 days after therapy qualified as cure. The mice were humanely sacrificed when becoming moribund, or when their tumors reached 1000 mm^3^ (alternatively earlier if becoming ulcerated). The sacrifices were performed following the standard operating procedure. Each treatment group consisted of eight mice.

### 2.4. Surface Heat Shock Protein 70 (HSP70) Measurement

After in vitro PDT protocol on cells treated with follow-up 1 h incubation in complete growth medium, exposure to edelfosine (1 h), or their combination, SCCVII cells were incubated 20 min on ice with FITC-conjugated mouse antiHSP70 monoclonal antibody (SPA-810F1, Stressgen Biotechnologies, Victoria, BC, Canada) or its isotype control FITC-conjugated ChromPure Mouse IgG (Jackson ImmunoResearch Laboratories, Inc, (West Grove, PA, USA). Next, the cells were detached from the monolayer using a rubber policeman and processed by flow cytometry. Fluorescence values obtained with the isotype control antibody were used for the background subtraction. Flow cytometry analysis of cell surface-exposed HSP70 was performed on a Coulter Epics Elite ESP (Coulter Electronics, Hialeah, FL, USA) with cells in advanced necrosis and cell fragments excluded on their light scatter signals. The results were based on the FITC fluorescence values in arbitrary units per cell, with 2 × 10^4^ cells included into each test.

### 2.5. Statistical Analysis

The Mann–Whitney test was used for flow cytometry data analysis and the results were presented as mean ± SD. To accommodate the confidence intervals for recognizing the difference between separate experimental groups at 95%, the threshold for statistical difference was set at 5%. The Log-Rank test-based survival analysis was applied with tumor response/cure results, with data plotted as Kaplan–Meier curves. The threshold for statistical significance was set also to 5%. This methodology functions by testing the null hypothesis (no difference between experimental groups in the probability of an event). It is a nonparametric test that compares estimates of the hazard functions of the two groups at each of observed event time (from day zero to day 90 after PDT; summarized for all time points where there is an event).

## 3. Results

To obtain an indication on whether combining PDT with edelfosine treatment could be therapeutically beneficial, it was first examined how their joint usage affects the expression of HSP70 on the surface of tumor cells. This heat shock protein is the major stress-inducible member in this large protein family, and its expression on the cell surface functions as danger signal, alerting the host of inflicted cell injury that can be recognized by immune cells bearing Toll-like receptors [25,26]. The application of PDT and exposure to edelfosine are both known to prompt the appearance of HSP70 on the surface of treated cells, and this has been attributed to their induction of ER stress and unfolded protein response (UPR) signaling [25,27]. The translocation of this heat shock protein into the plasma membrane (connected with its role of danger signal and DAMP (damage-associated molecular pattern)) is the recognized sign of ER-centered cellular stress [26] and the extent of its surface expression reflects the intensity of stress insult sustained by the cells.

Hence, SCCVII tumor cells were treated with Photofrin-based PDT, exposure to edelfosine, or both these treatments, and the surface HSP70 expression on these cells one hour post-treatment were assessed using the flow cytometry technique described in our previous work [25]. The chosen PDT treatment was found to kill >80% of SCCVII cells, while the edelfosine treatment (25 μg/mL for 1 h) had no impact on cell survival [25]. In conformance to the previous findings, the PDT-only treatment resulted in a notable upsurge of the surface HSP70 signal, while this was even more pronounced with edelfosine-alone exposure (Figure 1). Importantly, the detected surface HSP70 levels became elevated even higher after the combined treatment with PDT and edelfosine. Statistically, the expression levels of surface HSP70 with the combined treatment group were significantly higher than those of any other treatment group.

The impact of edelfosine on the response of SCCVII tumors growing in syngeneic C3H/HeN mice to PDT was then examined. The photosensitizing drug used for PDT was Photofrin, the classical porphyrin photosensitizer still largely used in clinical practice [1]. For the combined treatment, edelfosine (0.1 mg/mouse) was injected either intraperitoneally (i.p.) or peritumorally (p.t.) at 24 h either before or after PDT light.

The results reveal that the most effective tested protocol was with edelfosine injected p.t. 24 h post PDT. This treatment almost doubled the tumor cure rates from around 50% (obtained with PDT only) to close to 90% (Figure 2). Comparing this effect using local edelfosine administration in the tumor area with the systemic application (i.p.) of the same drug dose in the same time interval, it is evident that consequently reduced tumor concentrations obtained in the latter case were sufficient for attaining only a marginal therapeutic impact. On the other hand, edelfosine injected p.t. into mice bearing PDT-untreated SCCVII tumors showed no influence on tumor cures. The results also demonstrate that the edelfosine treatment applied 24 h before PDT light produced no evident therapeutic benefit. The same was found with edelfosine injected 1 h pre PDT.

Since it is known that the antitumor effects of edelfosine can vary depending on the specific details of metabolic activity of individual tumors, the therapeutic benefit of combining PDT and the treatment with this antitumor lipid was assessed with several other tumor models. In these further investigations, edelfosine administration was restricted to p.t., and the consequent reduction in the risk from systemic cytotoxic damage permitted elevation of the drug dose to 0.25 mg/kg (0.477 mmol/mouse).

Experiments in MCA205 fibrosarcomas growing on syngeneic C57BL/6 mice produced similar findings to those in SCCVVI carcinomas. The results show that the edelfosine treatment 24 h after PDT was again most effective, practically doubling the tumor cure rates compared to the PDT alone treatment (Figure 3). The difference in survival between these two treatment groups was statistically significant. In contrast, administering edelfosine immediately after PDT produced no evident improvement in tumor response. Similarly, applying edelfosine treatment one hour after PDT produced only marginal improvement in tumor cures, which was statistically not significant.

An important difference was obtained with Lewis lung carcinoma. With this tumor model, the edelfosine treatment given immediately after PDT proved highly effective, producing a statistically significant increase (approximately three-fold) in tumor cure rates in host mice (Figure 4). As with SCCVII and MCA205 tumors, the edelfosine-only treatment had no detectable effect on tumor cures with this tumor model.

## 4. Discussion

Edelfosine, a synthetic ether-linked analog of naturally occurring phosphatidylcholine and lysophosphatidylcholine, is one of the most investigated ether lipids [16]. This alkyl-lysophospholipid was first synthesized in 1969 by Günter Kny [28], with ether bonds in the C1 and C2 carbons of the glycerol backbone. The absence of ester bonds in its structure renders edelfosine inaccessible to degradation by lipases and this is responsible for its high metabolic stability in comparison to the natural counterparts [16]. Edelfosine was shown to be selectively internalized by cancer cells accumulating mainly in the ER while not targeting DNA, and compromising their survival (while sparing healthy cells) as it acts by interfering with multiple physiological pathways [29,30]. Edelfosine has also shown immune-modulating properties. It was reported to exhibit a strong anti-inflammatory activity, inhibit T cell proliferation, reduce the expression of MHC class II molecules, and elicit a Type I interferon response [31]. Pre-clinical and clinical studies with edelfosine have indicated its potential for treatment of various human solid tumors and leukemias [30,32]. In addition, edelfosine, due to the pleiotropic nature of its effects, has prospects for use in treating parasitic diseases, autoimmune conditions, and viruses like HIV-1 [16,29]. It is suitable for both oral and intravenous administration. Importantly, biodistribution studies have demonstrated a low retention of edelfosine in normal tissues and no significant systemic toxic side-effects were detected with this drug, including a lack of notable cardiotoxicity, hepatotoxicity, bone marrow toxicity, or renal toxic injury [33].

In the presented study, we demonstrate that edelfosine can serve as a potent adjuvant to PDT for the eradication of solid malignant tumors. The results with three different tumor models with immunocompetent syngeneic host mice showed consistently that edelfosine treatment has the potency for at least doubling the tumor cure rates attained by PDT only. It has also become clear that for the optimal timing of edelfosine treatment is that it needs to be delivered after PDT. The best results with SCCVII carcinomas and MCA205 fibrosarcomas were obtained with the injections of this drug 24 h after PDT. However, the evidence with Lewis lung carcinomas reveals that with some types of cancerous lesions, the administration of edelfosine immediately after PDT could be also highly effective. The cause cannot be ascribed to differences in mouse strain physiology, since MCA205 fibrosarcomas and Lewis lung carcinomas are growing in the same host strain of mice. Probably, the cause can be credited to malignant lesion-specific variations in the alteration of lipid metabolism and associated diversification in lipid composition of their membranes, as well as in the abundance and organization of lipid rafts present in these membranes [15].

It is also cogent that edelfosine treatment produced no beneficial therapeutic effect with the reverse order (given before PDT). The reason for this can have several conceivable explanations, offering bona fide determinants of the underlying mechanisms and/or their combinations: (i) it could come from the fact that the edelfosine-induced changes in lipid activity and cellular membrane structure are most destructive to cancer cells that have already sustained damage from PDT, (ii) the potentiation by edelfosine of PDT-inflicted membrane injury and obstruction of its repair, (iii) the interaction of edelfosine-triggered cell death signaling with PDT-triggered survival signaling pathways, and (iv) the blocking of PDT-induced SCAP/SREBP repair pathway-mediated lipogenesis and lipid homeostasis restoration by edelfosine-induced activity, including the downregulation of the main pathways for the removal of cholesterol and lipids from the cells. The results with surface HSP70 expression measurement (Figure 1) are consistent with the interpretation that the cellular stress inflicted by PDT becomes amplified by edelfosine treatment in a synergistic manner. Thus, edelfosine treatment-induced ER stress may override the PDT-induced survival signaling in the ER and switch its signaling pathways of cytoprotective nature towards cell death-promoting signal transduction. Obviously, it would be of a considerable benefit to have molecular/biological specificities of the interaction between tumor PDT and anti-tumor lipid activity more precisely characterized by further targeted investigations.

The highly critical importance of lipids in the response of solid tumors to PDT remains largely unrecognized and unappreciated. The fact that lipid hydroperoxides are by far the most abundant non-radical intermediates formed by PDT treatment also remains neglected [7]. Their reactions (lipid peroxidation) could alter surrounding biomolecules including proteins, other lipids, and nucleic acids, disseminating their damaging action to other (possibly more vital) sites. The scope of these propagative reactions remains largely unknown due to limitations in the methodology for monitoring the underlying events, but there are very strong indications that these secondary chain reactions could be far-reaching and relatively long-lasting due to the knowledge that the lifetimes of these peroxidative intermediates are far longer than those of singlet oxygen and free oxygen radicals formed during PDT treatment. On the other hand, there is an increased appreciation that stress signaling networks are determinants of PDT outcomes and a recognition of pivotal roles of lipids in these signal transduction pathways.

The present study featuring edelfosine can serve as a proof-in-principle argument for the use as PDT adjuvants other anti-cancer lipids optimized in recent advances of this field or developed in future studies. It appears that, generally, PDT responses in tumors are especially vulnerable to alterations in the lipid microenvironment and this can be highly effectively exploited for therapeutic benefit.

## Figures and Tables

**Figure 1 pharmaceutics-15-02723-f001:**
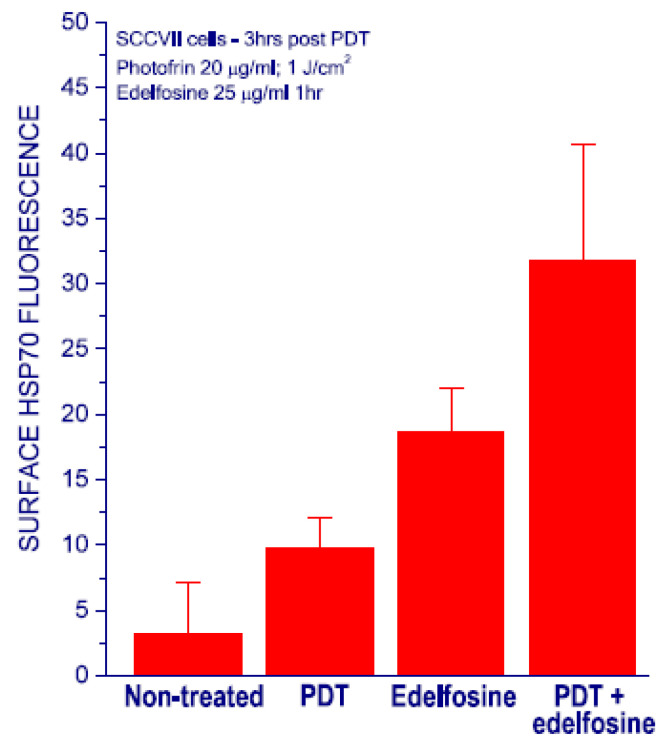
Expression of HSP70 on the surface of SCCVII tumor cells after Photofrin-PDT treatment, exposure to edelfosine, or their combination. For in vitro PDT treatment, the cells were incubated with Photofrin (20 μg/mL) for 24 h followed by illumination (1 J/cm^2^) with 630 ± 10 nm light, and then further post-incubation for 1 h in growth medium before harvesting for flow cytometry. For the edelfosine treatment, previously untreated or PDT-treated cells were exposed to the drug (25 μg/mL, i.e., 48 μmol/mL) for one hour under the full-growth condition. The expression of HSP70 on the cell surface was measured by flow cytometry using FITC-conjugated mouse anti-HSP70 monoclonal antibody (SPA-810F1). Bars are SD. Values in the PDT + edelfosine group were statistically higher than those in the other groups (*p* < 0.05). All the experiments were confirmed by at least one repeat experiment.

**Figure 2 pharmaceutics-15-02723-f002:**
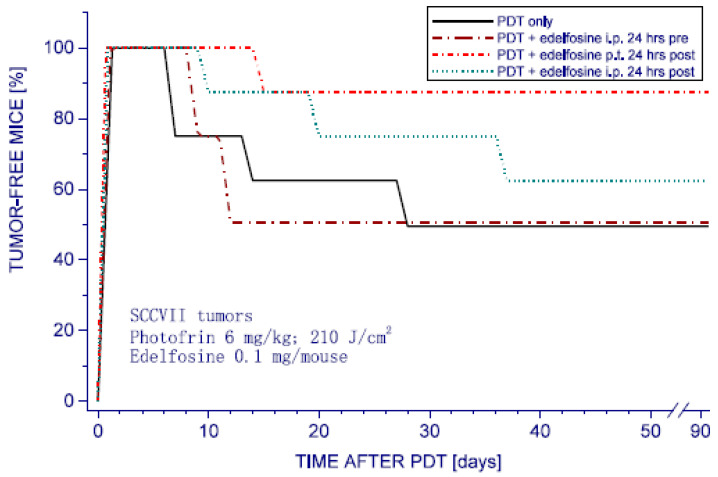
The impact of edelfosine treatment on PDT response of SCCVII tumors. Subcutaneous SCCVII tumors growing in C3H/HeN mice were PDT-treated by first injecting Photofrin (6 mg/kg i.v.) following exposure to 630 ± 10 nm light (210 J/cm^2^). 24 h later light (210 J/cm^2^). Edelfosine (0.1 mg/mouse, i.e., 0.194 mmol/mouse) was injected i.p. or p.t. at the indicated time before or after PDT light. “Tumor-free mice” in the ordinate are the mice with no palpable tumor. The response of the group with PDT plus edelfosine p.t. at 24 h post was statistically significantly different compared to PDT only and other groups (*p* < 0.05).

**Figure 3 pharmaceutics-15-02723-f003:**
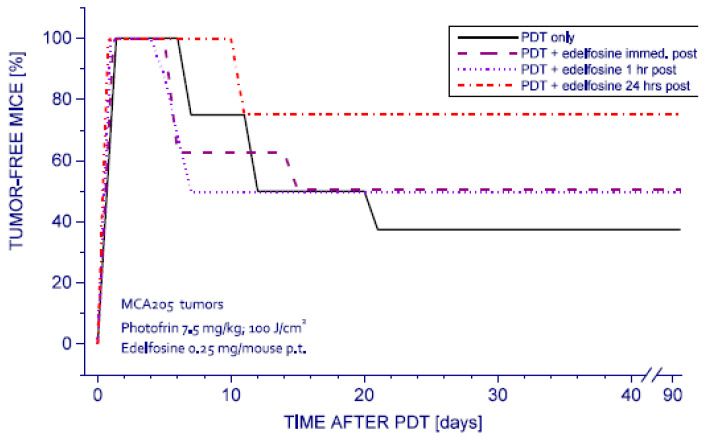
Response of MCA205 tumors to PDT and edelfosine. Subcutaneous MCA205 tumors growing in C57BL/6 mice were PDT-treated (Photofrin 7.5 mg/kg i.v. followed 24 h later by 630 ± 10 nm light (100 J/cm^2^)). Edelfosine (0.25 mg/mouse p.t.) was injected at indicated time after PDT light treatment. The response of the group with edelfosine following PDT 24 h later was statistically significantly different than PDT only and other groups (*p* < 0.05).

**Figure 4 pharmaceutics-15-02723-f004:**
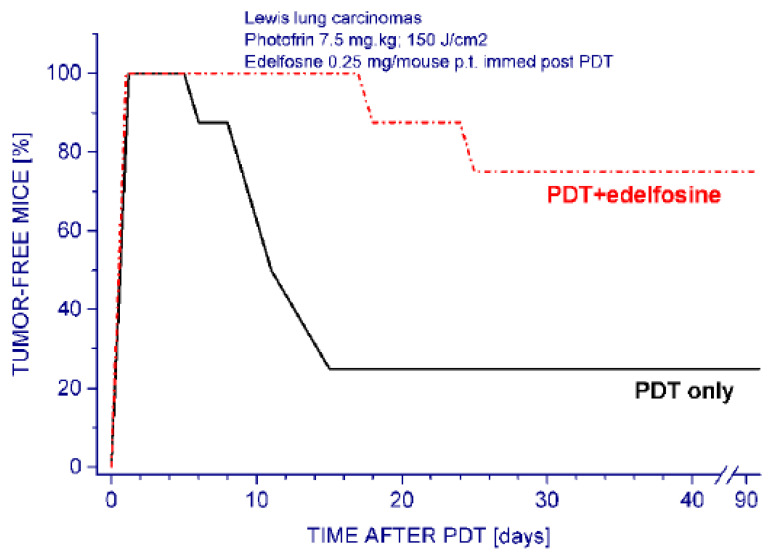
The response of Lewis lung tumors. to PDT and edelfosine. Subcutaneous tumors growing in C57BL/6 mice were PDT-treated (Photofrin 7.5 mg/kg i.v. followed 24 h later by 630 ± 10 nm light (150 J/cm^2^)). Edelfosine (0.25 mg/mouse p.t.) was injected immediately post PDT light treatment. The response of the group with PDT plus edelfosine was statistically significantly different than PDT-only group (*p* < 0.05).

## Data Availability

The data supporting reported results can be found in the author’s personal storage in British Columbia Cancer Research Centre.

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
