# Peer review of "Photodynamic Therapy Supported by Antitumor Lipids"

_pharmaceutics, 2023, doi:10.3390/pharmaceutics15122723_

Round 1

Reviewer 1 Report

Comments and Suggestions for Authors

Recognizing the crucial role of lipids in solid tumor response to photodynamic therapy (PDT) is paramount. The prevalence of lipid hydroperoxides, as the primary non-radical intermediates generated during PDT, warrants attention. Their reactions, leading to lipid peroxidation, can impact surrounding biomolecules, such as proteins, other lipids, and nucleic acids, extending their damaging effects to additional sites.

In the provided study, the author showcased the efficacy of edelfosine as a potent adjuvant to PDT in eliminating solid malignant tumors. Results from three distinct tumor models with immunocompetent syngeneic host mice consistently demonstrated that edelfosine treatment could double the tumor cure rates achieved by PDT alone. Hence, this research is invaluable and holds significant application potential for new, more effective therapy for solid tumors.

However, I have a few comments on the above manuscript:

1. Please discuss the aspect of anticancer lipids, their mechanism of action, and edelfosine itself in the Introduction section. Some of the above information is in the Results section. Still, such information should be included at the beginning of the article to introduce the topic to the reader who is not an expert.

2. In the Materials and Methods section, please indicate precisely what doses of edelfosine were used in in vitro tests.

3. Please supplement the article with in vitro tests on the MCA205 and Lewis lung carcinoma lines.

4. Please present a unified scheme of in vivo experiments performed, not different variants tested for different types of tumors. To compare the effects of the proposed therapeutic approach, please present the same tested variants in all tested tumor types.

5. Please correct the Y-axis signature in the graphs regarding in vivo studies because, according to the presented description, the mice were 100% tumor-free at the beginning of their observation.

Author Response

  1. As suggested by the reviewer, the paragraph describing edelfosine was moved from the Results section to the Introduction, and is now on page 3 lines 117-122. Basic information on anticancer lipids in general is provided in the Introduction on lines 88-116.
  2. The edelfosine dose used in vitro was 25 μg/ml, and this information is now included the Materials and Methods section (134-135) as requested.
  3. In vitro tests with edelfosine on the expression of HSP70 have already been done extensively with a great number of various tumor or other cells, and performed in combination with PDT, thermal or other stress treatments. The results were consistent with different cells as has been reported by various investigators. Our results with Lewis lung cells have also already been published. Based on this, there is no uncertainty of the compliance of the results of possible additional testing with MCA205 and Lewis lung cells with those presented with SCCVII cells (Figure 1). I am, therefore, against the suggested testing with additional cell lines for the reason of redundancy.
  4. The suggestion to use the same tested variants in all tested tumor types for a better comparison of the proposed therapeutic approach cannot be carried out. The first reason is the significant differences in the sensitivity to PDT of three tested tumor models. We therefore used comparable ranges of tumor cure rates for PDT only groups with different tumor models. This allowed us to adequately attest rises from moderate to high tumor cure rate levels. For edelfosine dose, we had to use a lower dose in comparing intraperitoneal and intratumoral delivery. Focusing solely on intratumoral route later allowed us to test increased (more effective) dose of edelfosine.
  5. The reviewer missed the detail that the mice were 0% tumor-free at the time of light treatment (day zero) but this state changed rapidly within one day and they became 100% tumor free at day 1. This rapid tumor disappearance (which may not be permanent but be followed by tumor recurrence later) is typical for PDT.

Reviewer 2 Report

Comments and Suggestions for Authors

This manuscript by Dr. Mladen Korbelik describes a combination of edelfosine, an anti-tumor lipid with photofrin-PDT using different mouse models. Tumor cure rates were significantly better when edelfosine was applied post-PDT, compared to pre-PDT.

The manuscript is acceptable in its current form except it’s not clear how many mice were used in Figures 2-4. After incorporating this information, the manuscript will be suitable for publication in Pharmaceutics.

Author Response

The information on the mouse numbers is now provided (lines 158-159).

Reviewer 3 Report

Comments and Suggestions for Authors

The manuscript written by M. Korbelik reports the tumor response to PDT with addition or not of edelfosine, an antitumor lipid. The studies have been conducted on three different tumoral cell lines grown in syngeneic mice. This study is well described. Here are some proposed improvements:

-          Korbelik describes the role of the different stakeholders in the response to treatment or the functioning of cells. Some explanations of HSP could also be added, for example in the Results paragraph.

-          p3, line 126, 129 and p4, line 159, p6, line 240: it would be better to add the amount of prepared and injected edelfosine in mmol/mL, as only this value (and not the one in mg/mL or µg/mL) can be compared with other from the literature.

-          p4, line 176: replace “PDT protocol followed by “ by “PDT protocol on cells treated with”

-          indicate in the figures if the experiments have been performed in duplicate or triplicate or else

-          p6, Fig 2: could the author explain more precisely the y axis? Explain what “tumor-free mice” indicates and how it is measured. Also indicate at which timepoint, the mice responses have been taken. Increase font size of the legend.

Comments on the Quality of English Language

English language is of good quality, some minor errors can be found.

Author Response

  1. As suggested by the reviewer, basic explanation on HSP is now inserted (lines 213-216).
  2. Edelfosine concentration/doses used are now listed also in molar values (lines 133,134, 137, 258, 267, and 273.
  3. The text “PDT protocol followed by…” was corrected as suggested (now on line 186).
  4. The statement “All the experiments have been confirmed by at least one repeat experiment” is now included (lines 261-262).
  5. It is explained that “tumor-free mice” in the ordinates are the mice with no palpable tumor (line 274). The mice were examined every second day after PDT for 90 days and the presence or absence of tumor on each of them was recorded (lines 179-180). The font size of the legend in Figure 2 was increased.

Round 2

Reviewer 1 Report

Comments and Suggestions for Authors

I accept an article in the present form.